# The Journey, Not the Destination:
# How Data Guides Diffusion Models

## Abstract

Diffusion models trained on large datasets can synthesize photo-realistic images of remarkable quality and diversity. However, *attributing* these images back to the training data—that is, identifying specific training examples which *caused* an image to be generated—remains a challenge. In this paper, we propose a framework that: (i) provides a formal notion of data attribution in the context of diffusion models, and (ii) allows us to *counterfactually* validate such attributions. Then, we provide a method, Journey-TRAK, for computing these attributions efficiently. Finally, we apply Journey-TRAK to find (and evaluate) such attributions for denoising diffusion probabilistic models trained on CIFAR-10 and latent diffusion models trained on MS COCO. We provide code at this https URL.

## 1 Introduction

Diffusion models can generate novel images that are simultaneously photorealistic and highly controllable via textual prompting (Ramesh et al., 2022; Rombach et al., 2022). A key driver of diffusion models' performance is training them on massive amounts of data (Schuhmann et al., 2022). Yet, this dependence on data has given rise to concerns about how diffusion models use it.

For example, Carlini et al. (2021); Somepalli et al. (2022) show that diffusion models often memorize training images and "regurgitate" them during generation. However, beyond such cases of direct memorization, we currently lack a method for *attributing* generated images back to the most influential training examples—that is, identifying examples that *caused* a given image to be generated. Indeed, such a primitive—a *data attribution method*—would have a number of applications. For example, previous work has shown that attributing model outputs back to data can be important for debugging model behavior (Shah et al., 2022), detecting poisoned or mislabelled data (Lin et al., 2022), and curating higher quality training datasets (Khanna et al., 2019). Within the context of diffusion models, data attribution can also help detect cases of data leakage (i.e., privacy violations), and more broadly, can be a valuable tool in the context of tracing content provenance relevant to questions of copyright (Andersen et al., 2023; Images, 2023). Finally, synthetic images generated by diffusion models are now increasingly used across the entire machine learning pipeline, including training (Azizi et al., 2023) and model evaluation (Kattakinda et al., 2022; Wiles et al., 2022; Vendrow et al., 2023). Thus, it is critical to identify (and mitigate) failure modes of these models that stem from training data, such as bias propagation (Luccioni et al., 2023; Perera & Patel, 2023) and memorization. Motivated by all the above needs, we thus ask:

*How can we reliably attribute images synthesized by diffusion models back to the training data?*

Although data attribution has been extensively studied in the context of *supervised* learning (Koh & Liang, 2017; Ghorbani et al., 2019; Jia et al., 2019; Ilyas et al., 2022; Hammoudeh & Lowd, 2022; Park et al., 2023), the generative setting poses new challenges. First, it is unclear *what particular behavior* of these models we hope to attribute. For example, given a generated image, certain training images might be responsible for the look of the background, while others might be responsible for the choice of an object appearing in the foreground. Second, it is not immediately obvious how to *verify* the attributions. In supervised settings, a standard approach is to compare the outputs of the original model on given inputs with those of a new model

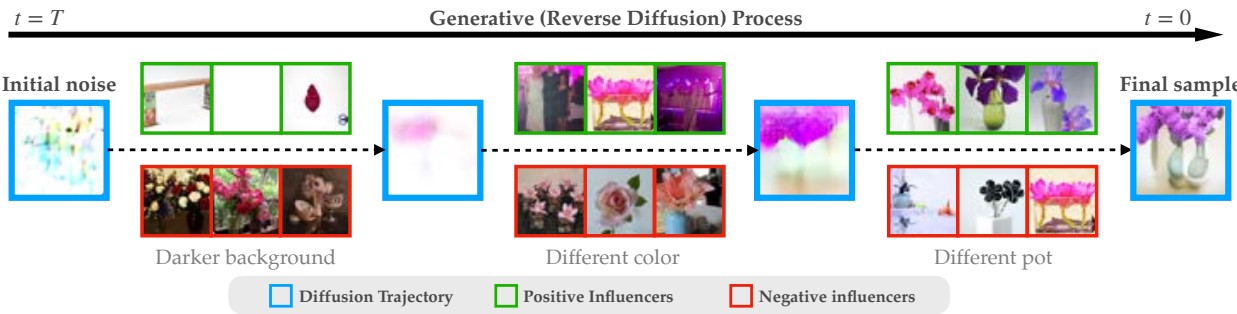

Figure 1: **Overview of our attribution framework.** For a given synthesized image, we apply Journey-TRAK, our attribution method, at individual steps along the diffusion trajectory. At each step $t$, Journey-TRAK pinpoints the training examples with the highest influence (positive in green, negative in red) on the generative process at that step. In particular, positive influencers guide the trajectory towards the final sample, while negative influencers guide the trajectory away from it. We observe that negative influencers increasingly resemble the final sample (the grey text highlights notable differences with the final sample). For more examples, see Appendix E.

trained on a new dataset after removing the attributed examples. However, in the generative setting it is less clear how to make such comparisons.

**Our contributions.** In this work, we present a data attribution framework for diffusion models. This framework reflects, and is motivated by, the fact that diffusion models iteratively denoise an initial random seed to generate the final image. In particular, rather than attributing *only* the final generated image, i.e., the "destination," we attribute each individual step along the (denoising) "journey" taken by diffusion model (see Figure 1). This approach shifts our focus from the specific final image to the *distribution* of possible generated images and, in particular, how this distribution evolves across the diffusion process. As we demonstrate, this framework also enables us to attribute specific *features* of the final generated image.

To analyze this framework, we introduce two complementary metrics for evaluating the resulting attributions based on their counterfactual impact on the distribution of generated images (rather than on specific samples). Finally, we provide an efficient method for computing such attributions, building on data attribution approaches developed for the supervised setting (Ilyas et al., 2022; Park et al., 2023). We then apply our method Journey-TRAK to denoising diffusion probabilistic models (DDPM) (Ho et al., 2020) trained on CIFAR-10 (Krizhevsky, 2009), and latent diffusion models (LDM) (Rombach et al., 2022) trained on MS COCO (Lin et al., 2014). In both of these settings, we obtain attributions that are validated by our metrics and also visually interpretable.

## 2 Preliminaries

We first provide background on data attribution. Then, we give a brief overview of diffusion models, highlighting the components that we will need to formalize attribution for these models.

### 2.1 Data attribution

Broadly, the goal of training data attribution (Koh & Liang, 2017; Ilyas et al., 2022; Hammoudeh & Lowd, 2022; Park et al., 2023) is to trace model outputs back to the training data. Intuitively, we want to estimate how the presence of each example in the training set impacts a given model output of interest (e.g., the loss of a classifier) on a specific input.

To formalize this, consider a learning algorithm $\mathcal{A}$ (e.g., a training recipe for a model), together with an input space $\mathcal{Z}$ and a training dataset $S = (z_1, \ldots, z_n) \in \mathcal{Z}^n$ of $n$ datapoints from that input space. Given a datapoint $z \in \mathcal{Z}$, we represent the model output via a *model output function* $f(z, \theta(S)) : \mathcal{Z} \times \mathbb{R}^d \to \mathbb{R}$, where

$\theta(S) \in \mathbb{R}^d$ denotes the model parameters resulting from running algorithm $\mathcal{A}$ on the dataset $\mathcal{S}$. For example, $f(z, \theta(S))$ is the loss on a test sample $z$ of a classifier trained on $S$. ( Our notation here reflects the fact that the parameters are a function of the training dataset $S$.) We now define a *data attribution method* as a function $\tau \colon \mathcal{Z} \times \mathcal{Z}^n \to \mathbb{R}^n$ that assigns a score $\tau(z, S)_i \in \mathbb{R}$ to each training example $z_i \in S$.[1] Intuitively, we want $\tau(z, S)_i$ to capture the change in the model output function $f(z, \theta(S))$ induced by adding $z_i$ to the training set.

More generally, these scores should help us make *counterfactual* predictions about the model behavior resulting from training on an arbitrary subset $S' \subseteq S$ of the training datapoints. We can formalize this goal using the *datamodeling* task (Ilyas et al., 2022): given an arbitrary subset $S' \subseteq S$ of the training set, the task is to predict the resulting model output $f(z, \theta(S'))$. A simple method to use the attribution scores for this task, then, is to consider a *linear* predictor: $f(z, \theta(S')) \approx \sum_{i : z_i \in S'} \tau(z, S)_i$.[2]

This view of the data attribution as a prediction task motivates a natural metric for evaluating attribution methods: the agreement between the true output $f(z, \theta(S'))$ and the output predicted by the attribution method $\tau$. Park et al. (2023) consider the rank correlation between the true and predicted values of $f(z, \theta(S'))$ over different random samples $S' \subseteq S$ and name the corresponding metric the *linear datamodeling score*—we will adapt it to our setting in Section 3.

**Estimating attribution scores (efficiently).**   Given the model output function $f$ evaluated at input $z$, a natural way to assign an attribution score $\tau(z)_i$ for a training datapoint $z_i$ is to consider the *marginal* effect of including that particular example on the model output, i.e., have $\tau(z)_i = f(z, \theta(S)) - f(z, \theta(S \setminus \{z_i\}))$. We can further approximate this difference by decomposing it as:

$$\tau(z)_i = \underbrace{(\theta - \theta_{-i})}_{\text{(i) change in model parameters}} \cdot \overbrace{\nabla_\theta f(z, \theta)}^{\text{(ii) change in model output}}, \tag{1}$$

where $\theta_{-i}$ denotes $\theta(S \setminus \{i\})$ (Wojnowicz et al., 2016; Koh & Liang, 2017). We can compute the second component efficiently, as this only requires taking the gradient of the model output function with respect to the parameters; in contrast, computing the first component is not always straightforward. In simpler settings, such as linear regression, we can compute the first component explicitly, as there exists a closed-form solution for computing the parameters $\theta(S')$ as a function of the training set $S'$. However, in modern, non-convex settings, estimating this component efficiently (i.e., without re-training the model) is challenging. Indeed, prior works such as influence functions (Koh & Liang, 2017) and TracIn (Pruthi et al., 2020) estimate the change in model parameters using different heuristics, but these approaches can be inaccurate in such settings.

To address these challenges, TRAK (Park et al., 2023) observed that for deep neural networks, approximating the original model with a model that is *linear* in its parameters, and averaging the estimates over multiple $\theta$'s (to overcome stochasticity in training) yields highly accurate attribution scores. The linearization is motivated by the observation that at small learning rates, the trajectory of gradient descent on the original neural network is well approximated by that of a corresponding linear model (Long, 2021; Wei et al., 2022; Malladi et al., 2022). In this paper, we will leverage the TRAK framework towards attributing diffusion models.

## 2.2   Diffusion models

**Training and sampling from diffusion models.**   At a high level, diffusion models (and generative models, more broadly) learn a distribution $p_\theta(\cdot)$ meant to approximate a target distribution $q_{data}(\cdot)$ of interest (e.g., natural images). To perform such learning, given a (training) sample $\mathbf{x}_0 \sim q_{\text{data}}(\cdot)$, diffusion models first apply a stochastic *diffusion process* that gradually corrupts $\mathbf{x}_0$ by adding more noise to it at each step. This results in a sequence of intermediate latents $\{\mathbf{x}_t\}_{t \in [T]}$ sampled according to $\mathbf{x}_t \sim \mathcal{N}(\alpha_t \cdot \mathbf{x}_{t-1}, (1 - \alpha_t) \cdot I)$ where $\{\alpha_t\}_t$ are parameters of the diffusion process (Sohl-Dickstein et al., 2015; Song & Ermon, 2019; Ho

---

[1]Following the literature, we say that an example $z_i$ has a *positive (respectively, negative) influence* if $\tau(z, S)_i > 0$ (respectively, $\tau(z, S)_i < 0$).

[2]Similarly to the prior work (Park et al., 2023), we only consider linear predictors here.

| $t = 800$ | $t = 600$ | $t = 400$ | $t = 200$ | |
|---|---|---|---|---|

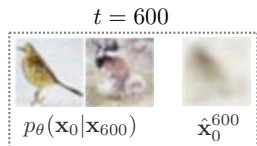 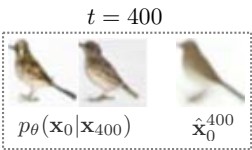 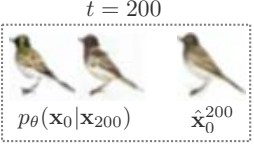 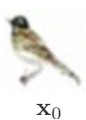

$p_\theta(\mathbf{x}_0|\mathbf{x}_{800})$ $\quad$ $\hat{\mathbf{x}}_0^{800}$ $\qquad$ $p_\theta(\mathbf{x}_0|\mathbf{x}_{600})$ $\quad$ $\hat{\mathbf{x}}_0^{600}$ $\qquad$ $p_\theta(\mathbf{x}_0|\mathbf{x}_{400})$ $\quad$ $\hat{\mathbf{x}}_0^{400}$ $\qquad$ $p_\theta(\mathbf{x}_0|\mathbf{x}_{200})$ $\quad$ $\hat{\mathbf{x}}_0^{200}$ $\qquad$ $\mathbf{x}_0$

Figure 2: **Samples from a diffusion trajectory.** We show samples from $p_\theta(\cdot|\mathbf{x}_t)$, i.e., the distribution of final images $\mathbf{x}_0$ conditioned on initializing from the latent $\mathbf{x}_t$ at step $t$, and the corresponding approximation $\hat{\mathbf{x}}_0^t$ (a proxy for the expectation of this distribution, i.e., $\mathbb{E}_{\mathbf{x}_0 \sim p_\theta(\cdot|\mathbf{x}_t)}[\mathbf{x}_0]$) for different values of $t$, together with the final generated image $\mathbf{x}_0$.

et al., 2020). Then, based on such sequences of intermediate latents, diffusion models learn a "denoising" neural network $\boldsymbol{\varepsilon_\theta}$ that attempts to run the diffusion process "in reverse."

Once such a diffusion model is trained, one can sample from it by providing that model with an initial seed $\mathbf{x}_T \sim \mathcal{N}(0, 1)$ (i.e., just a sample of random noise), and then applying the (trained) denoising network iteratively at each step $t$ (from $t = T$ to $t = 0$) to sample the corresponding *diffusion trajectory* $\{\mathbf{x}_t\}_{t \in [T]}$, ultimately leading to a final sample $\mathbf{x}_0 \sim p_\theta(\cdot) \approx q_{data}(\cdot)$.

**Conditioning sampling on partially denoised images.** Importantly, in this work, it will be also useful to consider the process of sampling a final image $\mathbf{x}_0$ when "resuming" the diffusion process after running it up to some step $t$—this is equivalent to continuing that process at step $t$ from the corresponding intermediate latent $\mathbf{x}_t$. We denote the distribution arising from sampling an image $\mathbf{x}_0$ when conditioning on the latent $\mathbf{x}_t$ by $p_\theta(\cdot|\mathbf{x}_t)$. Also, it turns out that we can approximate the multi-step denoising process of generating samples from $p_\theta(\cdot|\mathbf{x}_t)$ in a single step with the formula $\hat{\mathbf{x}}_0^t := c_1(\alpha_t) \cdot (\mathbf{x}_t - c_2(\alpha_t \cdot \boldsymbol{\varepsilon}_\theta(\mathbf{x}_t, t)))$, for some constants $c_1(\cdot), c_2(\cdot)$ that depend on the diffusion parameters $\{\alpha_t\}_t$ (Ho et al., 2020). In fact, $\hat{\mathbf{x}}_0^t$ is a proxy for the conditional expectation $\mathbb{E}_{\mathbf{x}_0 \sim p_\theta(\cdot|\mathbf{x}_t)}[\mathbf{x}_0]$, and under certain conditions $\hat{\mathbf{x}}_0^t$ is precisely equivalent to this expectation (Song et al., 2023; Daras et al., 2023).[3] See Figure 2 for an illustration of $p_\theta(\cdot|\mathbf{x}_t)$ and $\hat{\mathbf{x}}_0^t$ for different values of $t$.

**Types of diffusion models.** Finally, Denoising Diffusion Probabilistic Models (DDPMs) are a popular instantiation of diffusion models (Ho et al., 2020). More recently, Rombach et al. (2022) proposed a new class of diffusion models called latent diffusion models (LDMs), which perform the above stochastic process in the latent space of a pretrained encoder network. Moreover, Song et al. (2021); Ho & Salimans (2022) show that one can also *condition* diffusion models on some additional information, e.g. a text prompt. This way, one can control the semantics of the generated images by specifying such a text prompt. In this work, we will instantiate our data attribution framework on both unconditional DDPMs and conditional LDMs.

## 3 A Data Attribution Framework for Diffusion Models

In this section, we introduce our framework for attributing samples generated by diffusion models back to their training data. To this end, we will specify both *what* to attribute as well as how to *verify* the attributions. Specifically, in Section 3.1 we define data attribution for diffusion models as the task of understanding how training data influence the *distribution* over the final images at each step of the diffusion process. Then, in Section 3.2, we describe how to evaluate and verify such attributions.

### 3.1 Attributing the diffusion process step by step

Diffusion models generate images via a *multi-step* process. We thus decompose the task of attributing a final synthesized image into a corresponding series of stages, with each stage providing attributions for a single step of the diffusion process. This stage-wise decomposition allows for:

---

[3]This equivalence is referred to as the *consistency* property.

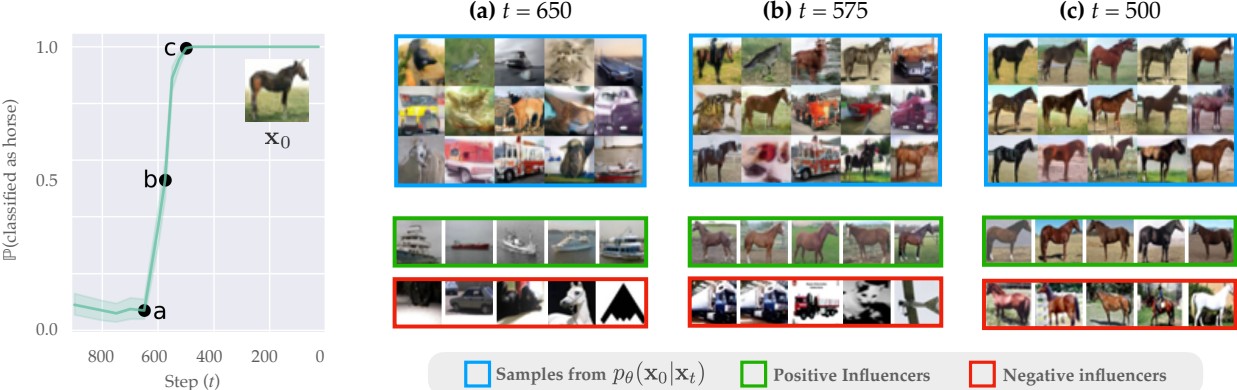

Figure 3: **Specific features appearing at specific steps. (Left)** For a given image of a horse ($\mathbf{x}_0$) generated by a CIFAR-10 DDPM model, we plot the likelihood that samples from the distribution $p_\theta(\cdot|\mathbf{x}_t)$ (see Section 2.2) are classified as a horse according to a CIFAR-10 classifier. This likelihood increases rapidly around steps 650 to 500, suggesting that these steps are most responsible for the formation of this feature. (**Top**) For three steps $t$ in this range, we visualize samples from $p_\theta(\cdot|\mathbf{x}_t)$. (**Bottom**) At each of these steps, we also visualize the training examples with the highest influence (positive in green, negative in red) identified by Journey-TRAK. Note that once the "horse" feature begins to appear (around $t = 575$), positive influencers begin to reflect it; and after this feature is "decided" (around $t = 500$), negative influencers *also* do so.

- **Fine-grained analysis.** Identifying influential training examples at each individual step gives us a fine-grained understanding of how data "guides" the diffusion process. This, in turn, allows us to capture, for example, that in some cases the same training example might be positively influential at early steps but negatively influential at later steps (see Appendix B.2).

- **Computational feasibility.** Computing gradients through a single step requires only a single backwards pass. So, it becomes feasible to apply existing efficient data attribution methods (Park et al., 2023; Pruthi et al., 2020) that involve computing gradients.

- **Feature-level attribution.** As we demonstrate below, features tend to form only within a small number of steps of the diffusion process. Thus, attributing at an individual step level allows us to isolate influences of training points on formation of specific features within the final generated image.

It remains now to define what exactly to attribute to the training data at each step. To this end, we first motivate studying the conditional distribution $p_\theta(\cdot|\mathbf{x}_t)$ (see Section 2.2) as a way to quantify the impact of each step $t$ of the diffusion process to the final sample $\mathbf{x}_0$. Next, we highlight how analyzing the evolution of this distribution over steps $t$ can connect individual steps to specific features of interest. Finally, building on these observations, we formalize our framework as attributing properties of this distribution $p_\theta(\cdot|\mathbf{x}_t)$ at each step $t$ to the training data.

**Studying the distribution $p_\theta(\cdot|\mathbf{x}_t)$.** At a given step $t$ of the generative process, the relevant information about the process up to that point is contained in the latent $\mathbf{x}_t$. While $\mathbf{x}_t$ itself might not correspond to a natural image, we can use it to directly sample from $p_\theta(\cdot|\mathbf{x}_t)$, i.e., the distribution of possible final images $\mathbf{x}_0$ when resuming the diffusion process at step $t$ with latent $\mathbf{x}_t$. When $t = T$, this distribution is precisely the diffusion model's learned distribution $p_\theta(\cdot)$, and at $t = 0$ it is simply the final sampled image $\mathbf{x}_0$. So, intuitively, the progression of this conditional distribution over steps $t$ informs us how the model gradually "narrows down" the possible distribution of samples to generate the final sample $\mathbf{x}_0$ (see Figure 2 for an illustration). A natural way to understand (and attribute) the impact of applying the diffusion process at each step $t$ on the final image $\mathbf{x}_0$ is thus to understand how this conditional distribution $p_\theta(\cdot|\mathbf{x}_t)$ evolves over steps.

**Connecting features to specific steps.** Given a final generated image, there might be many possible *features* of interest within this image. For example, for $\mathbf{x}_0$ in Figure 2, we might ask: *Why is there a grey bird? Why is the background white?* How can we quantify the impact of a particular step $t$ on a given feature in the final image? To answer this question, we simply sample from the conditional distribution $p_\theta(\cdot|\mathbf{x}_t)$ and measure the fraction of samples that contain the feature of interest. Now, if we treat this (empirical) likelihood as a function of $t$, the steps at which there is the largest increase in (i.e., the steepest slope of) likelihood are most responsible for the presence of this feature in the final image.

In fact, it turns out that such rapid increase in likelihood often happens within only a small interval; we observe this phenomenon for both small-scale unconditional models (DDPM trained on CIFAR-10, Figure 3) and large-scale text-conditional models (Stable Diffusion v2 trained on LAION-5B, Appendix B.3). As a result, we are able to tie the presence of a given feature in the final image back to a small interval of steps $t$ in the sampling process. This "phase transition" phenomenon has been observed and studied in concurrent work (Li & Chen, 2024; Sclocchi et al., 2024). In Figure E.4, we further explore this phenomenon for both different generated images and classifiers.

**Implementing our approach.** To implement our step-by-step attribution approach, we need a model output function (see Section 2.2) that is specific to a step $t$. As we motivated above, this function should be applied to samples from the conditional distribution $p_\theta(\cdot|\mathbf{x}_t)$. To that end, we introduce a step-specific model output function $f_t(p_{\theta(S)}(\cdot|\mathbf{x}_t), \theta(S))$. The function $f_t$ is intended to measure properties of the distribution $p_{\theta(S)}(\cdot|\mathbf{x}_t)$. For example, in Section 4 we define a concrete instantiation of $f_t$ that approximates the likelihood of the model to generate individual samples from $p_{\theta(S)}(\cdot|\mathbf{x}_t)$. Adapting the general definition of data attribution from Section 2.1, we can now define *data attribution for diffusion models* at a step $t$ as a function $\tau_t$ that assigns a score $\tau_t(\mathbf{x}_t, S)_i$ to each training example $z_i \in S$. This score indicates the change in $f_t(p_{\theta(S)}(\cdot|\mathbf{x}_t), \theta(S))$ induced by adding $z_i$ to $S$.

## 3.2 Validating data attribution for diffusion models

Visual inspection of the attributed training datapoints is a common heuristic for evaluating the quality of data attribution. However, visual similarity is not always reliable (Ilyas et al., 2022; Park et al., 2023). In particular, applications of data attribution such as data curation or model debugging often require that the attributions are *causally predictive*. Motivated by that, we evaluate attribution scores according to how accurately they reflect the corresponding training examples' *counterfactual* impact on the conditional distribution $p_\theta(\cdot|\mathbf{x}_t)$ using two different metrics, defined below.

**Linear datamodeling score.** The linear datamodeling score (LDS) is a measure of the effectiveness of a data attribution method that was introduced in Ilyas et al. (2022); Park et al. (2023) (see Section 2.1). This metric quantifies how well the attribution scores can predict the exact *magnitude* of change in model output induced by (random) variations in the training set. In our setting, we apply it to the step-specific model output function $f_t(p_{\theta(S)}(\cdot|\mathbf{x}_t), \theta(S))$. Specifically, we use the attribution scores $\tau$ to predict the diffusion-specific model output function $f_t(p_{\theta(S)}(\cdot|\mathbf{x}_t), \theta(S))$ as

$$g_\tau(p_{\theta(S)}(\cdot|\mathbf{x}_t), S'; S) := \sum_{i \,:\, z_i \in S'} \tau(\mathbf{x}_t, S)_i. \tag{2}$$

Then, we can measure the degree to which the predictions $g_\tau(p_{\theta(S)}(\cdot|\mathbf{x}_t), S'; S)$ are correlated with the true outputs $f_t(p_{\theta(S)}(\cdot|\mathbf{x}_t), \theta(S'))$ using the LDS:

$$LDS(\tau, \mathbf{x}_t) := \boldsymbol{\rho}(\{f_t(p_{\theta(S)}(\cdot|\mathbf{x}_t), \theta(S_j)) : j \in [m]\}, \{g_\tau(p_{\theta(S)}(\cdot|\mathbf{x}_t), S_j; S) : j \in [m]\}),$$

where $\{S_1, \ldots, S_m : S_i \subset S\}$ are randomly sampled subsets of the training set $S$ and $\boldsymbol{\rho}$ denotes Spearman's rank correlation (Spearman, 1904). To decrease the cost of computing LDS, we use $\widehat{\mathbf{x}}_0^t$ in lieu of samples from $p_{\theta(S)}(\cdot|\mathbf{x}_t)$ (see Section 2.2), since, as noted in Section 2.2, $\widehat{\mathbf{x}}_0^t$ turns out to be a good proxy for the the latter quantity. In other words, we consider $f_t$ and $g_\tau$ as functions of $\widehat{\mathbf{x}}_0^t$ rather than $p_{\theta(S)}(\cdot|\mathbf{x}_t)$.

**Retraining without the most influential images.** In practice, we may want to use the data attributions to intentionally steer the diffusion model's output. For example, we may want to remove all training examples that cause the resulting model to generate a particular style of images. To evaluate the usefulness of a given data attribution method in these contexts, we remove from the training set the most influential (i.e., highest scoring) images for a given target $\mathbf{x}_t$, retrain a new model $\theta'$, then measure the change in the conditional distribution $p_\theta(\cdot|\mathbf{x}_t)$ (see Section 2.2) when we replace $\theta$ with $\theta'$ only in the neighborhood of step $t$ in the reverse diffusion process. If the data attributions are accurate, we expect the conditional distribution to change significantly (as measured in our case using the FID distance for images (Heusel et al., 2017)). As we consider data attributions that are specific to each step, in principle we should use the denoising model *only* for the corresponding step $t$. However, the impact of a single step on the final distribution might be small, making it hard to measure. Hence, we assume that attributions change only gradually over steps and replace the denoising model for a *small interval* of steps (i.e., between steps $t$ and $t - \Delta$).

The first metric (LDS) is cheaper to evaluate, as we can re-use the same set of models to evaluate attributions for different target images and from different attribution methods. On the other hand, the latter metric more directly measures changes in the conditional distribution $p_\theta(\cdot|\mathbf{x}_t)$, so we do not need to rely on a specific choice of a model output function $f_t$.

## 4 Efficiently Computing Attributions for Diffusion Models

In this section, we describe how we can efficiently estimate data attributions for diffusion models using TRAK (Park et al., 2023). As we described in Section 2.1, we can decompose the task of computing data attribution scores into estimating two components: (i) the change in model parameters, and (ii) the induced change in model output. Following TRAK (Park et al., 2023), computing the first component (change in model parameters) only requires computing per-example gradients of the training loss (and in particular, does not require any re-training per each training datapoint). Similarly, computing the second component (change in model output) only requires computing gradients with respect to the model output function of choice (see Appendix D, as well as Section 3 of Park et al. (2023) for details). We now describe how we arrive at Journey-TRAK by adapting the estimation of the above two components to the diffusion model setting.

**Estimating the change in model parameters.** For diffusion models, the training process is much more complicated than the standard supervised settings (e.g., image classification) considered in Park et al. (2023). In particular, one challenge is that the diffusion model outputs a high-dimensional vector (an image) as opposed to a single scalar (e.g., a label). Even if we approximate the diffusion model as a *linear* model in parameters, naively applying TRAK would require keeping track of $p$ gradients for each training example (where $p$ is the number of pixels) and thus be computationally infeasible. Nonetheless, it is still the case that the presence of a single training example influences the optimization trajectory *only* via the gradient of the loss on that example—specifically, the MSE of the denoising objective. Hence, it suffices to keep track of a single gradient for each example. This observation allows us to estimate the change in model parameters using the same approach that TRAK uses (see Section 2.1).

An additional challenge is that the gradient updates in the diffusion process are highly stochastic due to the sampling of random noise. To mitigate this stochasticity, we average the training loss over multiple resampling of the noise at randomly chosen steps and compute gradients over this averaged loss.

**A model output function for diffusion models.** In Section 3, we motivated why we would like to attribute properties of the conditional distribution $p_{\theta(S)}(\cdot|\mathbf{x}_t)$, i.e., the distribution that arises from sampling when conditioning on an intermediate latent $\mathbf{x}_t$. Specifically, we would like to understand what training data causes the model to generate samples from this distribution. Then, one natural model output function $f_t$ would be to measure the likelihood of the model to generate these samples. Attributing with respect to such a choice of $f_t$ allows us to understand what training examples increase or decrease this likelihood.

In order to efficiently implement this model output function, we make two simplifications. First, sampling from $p_{\theta(S)}(\cdot|\mathbf{x}_t)$ can be computationally expensive, as this would involve repeatedly resampling parts of the diffusion trajectory. Specifically, sampling once from $p_{\theta(S)}(\cdot|\mathbf{x}_t)$ requires applying the diffusion model $t$

---

**Algorithm 1** Journey-TRAK, a data attribution method for diffusion models

---

1: **Input:** Model checkpoints $\{\theta_1^\star, ..., \theta_M^\star\}$, training dataset $S = \{\mathbf{z}_1, ..., \mathbf{z}_N\}$, target sequence $\{\mathbf{x}_1, ..., \mathbf{x}_T\}$ corresponding to $T$ steps, projection dimension $k \in \mathbb{N}$.
2: **Output:** Attribution scores $\tau(\mathbf{x}_t, S) \in \mathbb{R}^N$ for each $t$
3: $f_{\text{train}}(\mathbf{x}, \theta) \coloneqq \mathbb{E}_{\boldsymbol{\varepsilon}, t} \left\| \boldsymbol{\varepsilon} - \boldsymbol{\varepsilon}_{\theta(S)} \left( \sqrt{\bar{\alpha}_t}\mathbf{x} + \sqrt{1 - \bar{\alpha}_t}\boldsymbol{\varepsilon}, t \right) \right\|_2^2$           ▷ DDPM training loss
4: $f_t(\cdot, \theta)$ defined as in Equation (3)        ▷ Step-specific model output function $f_t$
5: **for** $m \in \{1, \ldots, M\}$ **do**
6:      $\mathbf{P} \sim \mathcal{N}(0, 1)^{p \times k}$            ▷ Sample random projection matrix
7:      **for** $i \in \{1, \ldots, N\}$ **do**
8:          $\phi_i \leftarrow \mathbf{P}^\top \nabla_\theta f_{\text{train}}(\mathbf{z}_i, \theta_m^\star)$       ▷ Compute training loss gradient at $\theta_m^\star$ and project
9:      **end for**
10:      **for** $t \in \{1, \ldots, T\}$ **do**
11:          $\widehat{\mathbf{x}}_0^{(t)} \leftarrow c_1(\alpha_t) \cdot (\mathbf{x}_t - c_2(\alpha_t \cdot \varepsilon_{\theta_m^\star}(\mathbf{x}_t, t)))$     ▷ Compute expectation of conditional distribution
12:          $g_i \leftarrow \mathbf{P}^\top \nabla_\theta f_t(\widehat{\mathbf{x}}_0^{(t)}, \theta_m^\star)$     ▷ Compute model output gradient at $\theta_m^\star$ and project
13:      **end for**
14:      $\Phi_m \leftarrow [\phi_1; \cdots; \phi_N]^\top$
15:      $G_m \leftarrow [g_1; \cdots; g_T]^\top$
16: **end for**
17: $[\tau(\mathbf{x}_1, S); \cdots; \tau(\mathbf{x}_T, S)] \leftarrow \frac{1}{m} \sum_{m=1}^M \Phi_m (\Phi_m^\top \Phi_m)^{-1} G_m$       ▷ Average scores over checkpoints
18: **return** $\{\tau(\mathbf{x}_t, S)\}$

---

times—in practice, $t$ can often be as large as 1000. Fortunately, as we described in Section 2.2, we can use the one-step estimate $\widehat{\mathbf{x}}_0^t$ as a proxy for samples from $p_{\theta(S)}(\cdot | \mathbf{x}_t)$, since it approximates this distribution's expectation $\mathbb{E}_{\mathbf{x}_0 \sim p_\theta(\cdot | \mathbf{x}_t)}[\mathbf{x}_0]$.

Second, it is computationally expensive to compute gradients with respect to the exact likelihood of generating an image. So, as a more tractable proxy for this likelihood, we measure the reconstruction loss[4] (i.e., how well the diffusion model is able to denoise a noisy image) when adding noise to $\widehat{\mathbf{x}}_0^t$ with magnitude matching the sampling process at step $t$. Specifically, we compute the Monte Carlo estimate

$$f_t \left( \widehat{\mathbf{x}}_0^t, \theta(S) \right) = \sum_{i=1}^k \left\| \boldsymbol{\varepsilon}_i - \boldsymbol{\varepsilon}_{\theta(S)} \left( \sqrt{\bar{\alpha}_t}\widehat{\mathbf{x}}_0^{(t)} + \sqrt{1 - \bar{\alpha}_t}\boldsymbol{\varepsilon}_i, t \right) \right\|_2^2, \tag{3}$$

where $\bar{\alpha}_t$ is the DDPM[5] variance schedule (Ho et al., 2020), $\boldsymbol{\varepsilon}_i \sim \mathcal{N}(0, 1)$ for all $i \in [k]$, and $k$ is the number of resampling rounds of the random noise $\boldsymbol{\varepsilon}$. Now that we have chosen our model output function, we can simply compute gradients with respect to this output to obtain the second component in Equation (1).

**The final algorithm.** We summarize our algorithm Journey-TRAK for computing attribution scores in Algorithm 1. We approximate the training loss (line 3) with different samples of noise $\varepsilon$ and step $t$. Note that to attribute a new target sequence, we only have to recompute lines 10-12.

**Comparison with** TRAK. Our method Journey-TRAK is inspired by TRAK, an attribution method for supervised settings. The main difference comes from the diffusion-specific model output function $f_t(\cdot, \theta)$. In particular, it is not immediately clear how to design such a function for the multi-step, sampling process in diffusion models in a way that is both computationally efficient and counterfactually predictive. We show that our choice for $f_t$ described in Equation (3) gives us an efficient proxy for samples from $p_{\theta(S)}(\cdot | \mathbf{x}_t)$, leading to a fast and effective data attribution method.

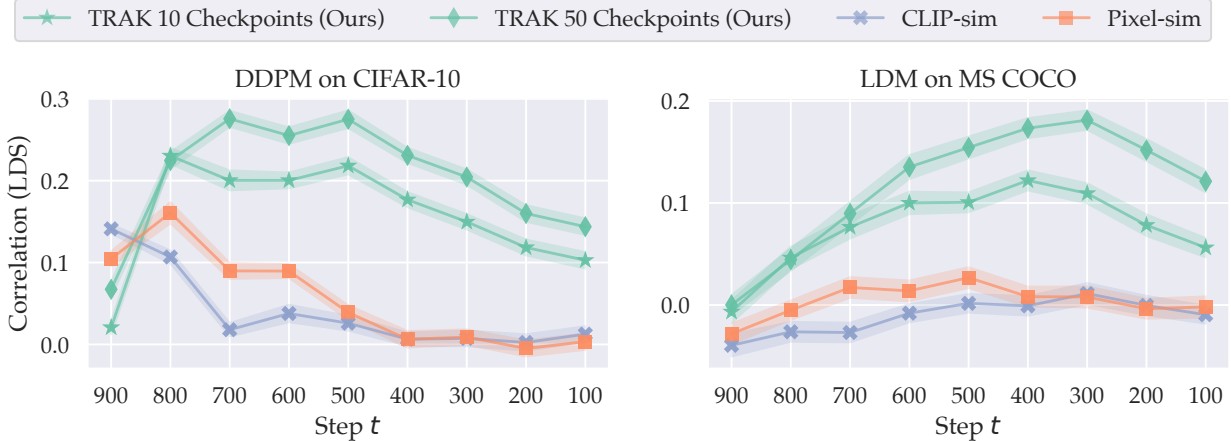

Figure 4: **Predicting model behavior.** The counterfactual predictiveness of attributions measured using the LDS score along the diffusion trajectory (at every 100 steps) for three different methods: Journey-TRAK (computed using 10 and 50 model checkpoints), CLIP similarity, and pixel similarity. Smaller steps are closer to the final sample. Shaded areas represent standard error.

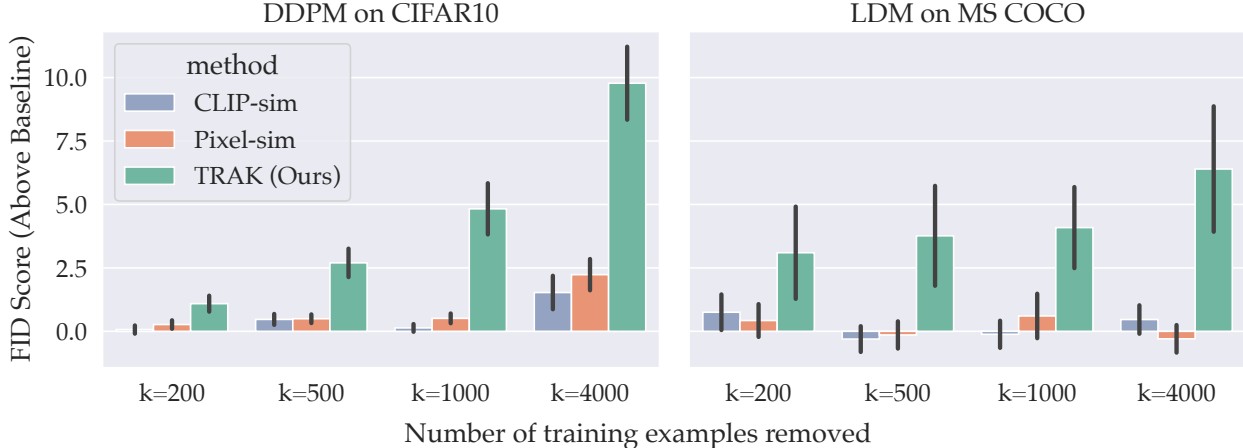

Figure 5: **Retraining without top influencers.** Change in the distribution of generated images $p_\theta(\cdot|\mathbf{x}_{400})$ when substituting the original model with a new model only between steps 400 and 300. The new model is trained without the $k$ top influencers of $\mathbf{x}_{400}$ according to attributions from Journey-TRAK (computed at step 400), CLIP similarity, or pixel similarity. To evaluate the change in distribution, we measure the increase in FID score over a baseline of models trained on the full dataset (see Section 3.2 for details). Bars represent standard error.

## 5 Experiments

To evaluate our data attribution method, we apply it to DDPMs trained on CIFAR-10 and LDMs trained on MS COCO. First, in Section 5.2, we visually inspect and interpet our attributions, and then in Section 5.3 we evaluate their counterfactual significance using the metrics we introduced in Section 3.2. In Section 5.4, we further explore how our data attributions can be localized to patches in pixel space. Finally, in Section 5.5, we investigate the value of our step-specific attributions for attributing the full diffusion trajectory.

---

[4]The reconstruction loss is a proxy for the likelihood of the generated image, as it is proportional to the evidence lower bound (ELBO) (Sohl-Dickstein et al., 2015; Song et al., 2023).

[5]We only consider DDPM schedulers in this work. The above derivation can be easily extended to other schedulers.

### 5.1 Experimental setup

We compute our data attribution scores using 100 DDPM checkpoints trained on CIFAR-10 and 50 LDM checkpoints trained MS COCO (see Appendix A for training details.). As baselines, we compare our attributions to two common image similarity metrics—CLIP similarity (i.e., cosine similarity in the CLIP embedding space) and cosine similarity in pixel space.

### 5.2 Qualitative analysis of attributions

In Figure 1, we visualize the sampling trajectory for an image generated by an MS COCO model, along with the most positive and negative influencers identified by Journey-TRAK (see Appendix E for additional visualizations of identified attributions on CIFAR-10 and MS COCO). We find that positive influencers tend to resemble the generated image throughout, while negative influencers tend to differ from the generated image along specific attributes (e.g., class, background, color) depending on the step. Interestingly, the negative influencers increasingly resemble the generated image towards the end of the diffusion trajectory.

Intuitively, we might expect that negative influencers would not resemble the final generated image, as they should to steer the trajectory away from that image. So, why do they in fact reflect features of the final generated image? To answer this question, we study the relationship between the top (positive and negative) influencers and the distribution $p_\theta(\cdot|\mathbf{x}_t)$ towards which we target our attributions. In Figure 3, for a given image of a horse generated by our CIFAR-10 DDPM, we plot the likelihood that images from $p_\theta(\cdot|\mathbf{x}_t)$ containing a horse (according to a classifier trained on CIFAR-10) as a function of the step $t$ (left). We also show the top and bottom influencers at three points along the trajectory (right). We find that the top influencers begin to reflect the feature of interest once the likelihood of this feature begins to grow. Yet, once the likelihood of the feature reaches near certain, the negative influencers *also* begin to reflect this feature. This behavior has the following intuitive explanation: after this point, it would be impossible to "steer" the trajectory away from presenting this feature. So, the negative influencers at later steps might now steer the trajectory away from other features of the final image (e.g., the color of horse) that has not yet been decided at that step. Additionally, images that do not reflect the "decided" features might no longer be relevant to steering the trajectory of the diffusion process.

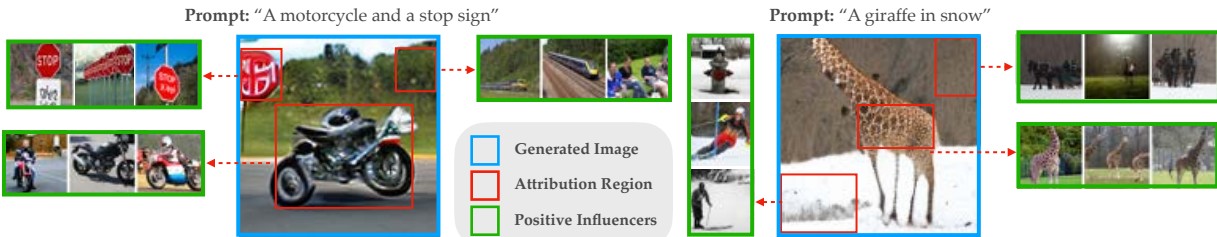

Figure 6: **Patch-based attribution.** We adapt Journey-TRAK to restrict attribution to user-specified patches of a generated image. We show examples of attributing patches capturing individual concepts in images synthesized by a latent diffusion model trained on MS COCO. Attributions are computed at step $t = 400$.

### 5.3 Counterfactually validating the attributions

We now evaluate our attributions using the metrics introduced in Section 3.2 to validate their counterfactual significance.

**LDS.** We sample 100 random 50% subsets of CIFAR-10 and MS COCO, and train five models per mask. Given a set of attribution scores, we then compute the Spearman rank correlation (Spearman, 1904) between the predicted model outputs $g_\tau(\cdot)$ (see Eq. (2)) on each training data subset according to the attributions and the (averaged) actual model outputs. To evaluate the counterfactual significance of our attributions over

the course of the diffusion trajectory, we measure LDS scores at every 100 steps over the 1000 step sampling process.

In Figure 4, we plot LDS scores for CIFAR-10 (left) and MS COCO (right) over a range of steps for our attribution scores as well as the two similarity baselines. Unlike in many computer vision settings (Zhang et al., 2018), we find that for CIFAR-10, similarity in pixel space achieves competitive performance, especially towards the start of the diffusion trajectory. However, for both CIFAR-10 and MS COCO, only Journey-TRAK is counterfactually predictive across the entire trajectory.

**Retraining without the most influential images.** We compute attribution scores on 50 samples from our CIFAR-10 and MS COCO models at step $t = 400$. Given the attribution scores for each sample, we then retrain the model after removing the corresponding top $k$ influencers for $k \in \{200, 500, 1000\}$. We sample 5000 images from two distributions: (1) the distribution arising from repeatedly initializing at $\mathbf{x}_{400}$ and sampling the final 400 steps from the original model; and (2) the distribution arising from repeating the above process but using the retrained model only for steps $t = 400$ to $t = 300$. We then compute FID distance between these distributions, and repeat this process for each sample at each value of $k$.

In Figure 5, we display the average FID scores (a measure of distance from the original model) after removing the $k$ most influential images for a given sample across possible values of $k$. We notice that, for all values of $k$, removing the top influencers identified by our attribution method has a greater impact than removing the most similar images according to CLIP or pixel space similarities.

### 5.4 Localizing our attributions to patches in pixel space

In Section 3, we discussed how step-by-step attribution allows us to attribute particular features appearing within a particular interval of steps. However, some features may appear together within a small interval, making it hard to isolate them only based on the step. Here we explore one possible approach for better isolating individual features: selecting a region of pixels (i.e., a *patch*) in a generated sample corresponding to a feature of interest, and restricting our model output function to this region. This way, we can restrict attributions only to the selected patch, which can be useful for understanding what caused a specific feature to appear (see Figure 6). To implement this model output function, we simply apply a pixel-wise binary mask to Equation (3) and ignore the output outside of the masked region. To test this approach, we generate images containing multiple features with an MS COCO-trained LDM. We then manually create per-feature masks for which we compute attribution scores with Journey-TRAK (see Figure 6). The resulting attributions for different masks surface training examples relevant *only* to the corresponding features in that region.

### 5.5 "Forgetting" how to generate an image

Our attribution scores and evaluation metrics are all step-specific. However, in practice we might care about identifying training images that impact the *full* diffusion pipeline. In particular, we might be interested in whether removing the important training images for a given synthesized image causes the diffusion model to "forget" how to generate this image.

Specifically, given a set of attribution scores for a synthesized image, we remove the top $k$ influencers (at step $t = 300$), retrain the model, and generate new images from scratch using the same random seed. Here, we leverage the fact that two diffusion models trained on the same dataset tend to generate similar images given the same random seed (Song et al., 2021); see Appendix B.1 for more details. We then compare the change in pixel space between the original and newly generated image. This process is distinct from our second evaluation metric, as (1) we directly compare two images rather than measure the distance between distributions, and (2) we re-generate images with our new model from scratch rather than restarting from some intermediate latent $\mathbf{x}_t$ and substituting the new model for only a small interval of steps (between $t$ and $t - \Delta$).

We perform this process for our attribution scores on CIFAR-10 as well as the two similarity baselines (see Figure 7). Our results suggest that Journey-TRAK is able to identify influential images that have a significant impact on the full diffusion trajectory of the diffusion model.

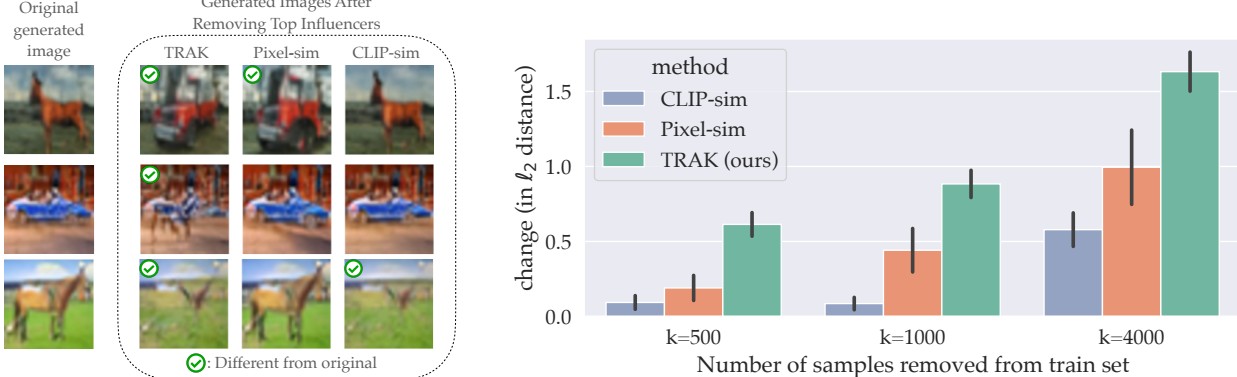

Figure 7: **"Forgetting" an image.** We quantify the impact of removing the highest scoring training examples according to Journey-TRAK, CLIP similarity, and pixel similarity (and re-training). **(Left)** We compare the original synthesized samples to those generated from the same random seed with the re-trained models. **(Right)** To quantify the impact of removing these images, we measure the $\ell_2$ distance between 60 synthesized samples and corresponding images generated by the re-trained models. Black bars represent standard error.

## 6 Related Work

While most prior works on data attribution focus on the supervised setting, some more recent works study attribution in generative settings, e.g. audio models Deng & Ma (2023). For diffusion models, recently Wang et al. (2023) propose a method for *efficiently evaluating* data attribution methods for generative models by creating custom datasets with known ground-truth attributions.

Concurrently to this work, Zheng et al. (2023) use Journey-TRAK to attribute diffusion models *across timesteps*, i.e., they provide global attributions; they rely on heuristic design choices to design $f_t$ . Lin et al. (2024) also propose a *global* attribution method; their method is based on Shapley values, a concept from economics; and Wang et al. (2024) propose a method based on approximate unlearning. Dai & Gifford (2023) employ ensembling techniques to attribute diffusion models, again on a global scale. We discuss additional related work in Appendix C.

## 7 Conclusion

In this work, we introduce a framework for data attribution for diffusion models and provide an efficient method for computing such attributions. In particular, we formalize data attribution in this setting as task of quantifying how individual training datapoints influences the distribution over final images *at each step* of the diffusion process. We demonstrate the efficacy of our approach on DDPMs trained on CIFAR-10 and LDMs trained on MS COCO. Our framework also constitutes a step towards better understanding of how training data influences diffusion models.

There are several directions for potential improvements and future work. First, our particular instantiation of the framework relies on proxies for the distribution $p_\theta(\cdot|x_t)$ of final generated images conditioned on a given step $t$, as well as for the likelihood of generating a given image. So, identifying more accurate proxies could help improve the quality of the resulting attributions. More broadly, we evaluate our framework on two academic-size datasets, but the most popular diffusion models (such as Stable Diffusion) are larger and trained on significantly larger datasets. Thus, while feasible in principle, scaling our framework to such settings is important. Finally, while we study the task of attributing individual steps, it would be valuable to perform data attribution for the full diffusion process.

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

# A    Experimental details

Throughout our paper, we train various diffusion models on CIFAR-10 and MS COCO.

**DDPM training on CIFAR-10.**    We train 100 DDPMs (Ho et al., 2020) on the CIFAR-10 dataset for 200 epochs using a cosine annealing learning rate schedule that starts at 1e-4. We used the DDPM architecture that match the original implementation (Ho et al., 2020), which can be found here `https://huggingface.co/google/ddpm-cifar10-32`. At inference time we sample using a DDPM scheduler with 50 inference steps.

**LDM training on MS COCO.**    We train 20 text-conditional latent diffusion models (LDMs) (Rombach et al., 2022) on the MS COCO dataset for 200 epochs using a cosine annealing learning rate schedule that ståarts at 2e-4. We use the exact CLIP and VAE used in Stable Diffusion v2, but use a custom (smaller) UNet, which we describe in our code. These models can be found here `https://huggingface.co/stabilityai/stable-diffusion-2-1`. At inference time, we sample using a DDPM scheduler with 1000 inference steps.

**LDS.**    We sample 100 random 50% subsets of CIFAR-10 and MS COCO, and train 5 models per mask. Given a set of attribution scores, we then compute the Spearman rank correlation (Spearman, 1904) between the predicted model outputs $g_\tau(\cdot)$ (see Eq. (2)) on each subset according to the attributions and the (averaged) actual model outputs. Because our model output and attributions are specific to a step, we compute LDS separately for each step. To evaluate the counterfactual significance of our attributions over the course of the diffusion trajectory, we measure LDS scores at each 100 steps over the 1000 step sampling process.

**Retraining without the most influential images.**    For our counterfactual evaluation in Section 5.3, we compute attribution scores on 50 samples from our CIFAR-10 and MS COCO models at step $t = 400$. Given the attribution scores for each sample, we then retrain the model after removing the corresponding top $k$ influencers for $k = 200, 500, 1000$. We compute FID based on 5000 images from each distribution, and repeat this process for each sample at each value of $k$.

**Journey-trak hyperparameters**    In the random projection step of Journey-TRAK, we use a projection dimension of $d = 4096$ for CIFAR-10 and $d = 16384$ for MS COCO. As in Park et al. (2023), we use multiple model checkpoints in order to compute the attribution scores. For CIFAR-10, we use 100 checkpoints, and for MS COCO, we use 20 checkpoints. In our code repository (github.com/MadryLab/journey-TRAK), we release the pre-computed Journey-TRAK features for all of our models, allowing for a quick computation of Journey-TRAK scores on new synthesized images. In Equation (3), we use $k = 20$ for both CIFAR-10 and MS COCO.

# B  Additional Analysis and Results

## B.1  Diffusion models are consistent across seeds

Song et al. (2021) show that in the limit of infinite capacity and training data, as well as perfect optimization, the embedding $\mathbf{x}_T$ obtained by diffusion models is *u*niquely identifiable. In other words, two independently trained diffusion models trained on the same dataset will embed an image $\mathbf{x}_0$ to the same embedding $\mathbf{x}_0$. They also provide empirical evidence that this phenomenon holds approximately in non-idealized settings.

Zhang et al. (2023) and Kadkhodaie et al. (2023) show empirically that the latent spaces of diffusion models we attribute are indeed highly aligned; i.e., two independently trained diffusion models will generate similar images given a shared embedding $\mathbf{x}_0$. We refer to this property as *seed consistency.*

Additionally, Khrulkov et al. (2022) provide an optimal transport perspective on this phenomenon.

The seed consistency property is critical for our attribution method Journey-TRAK, so we experimentally verify it. In fact, we find that images generated by many independently trained DDPMs on CIFAR-10 from the same random seed and nearly indistinguishable (see Figure B.1, right). To evaluate seed consistency quantitatively, we measure the $\ell_2$ distance between images generated by two models when using identical or distinct noise sequences, and find that matching the noise sequences leads to a far smaller $\ell_2$ distances (see Figure B.1, left).

We additionally evaluate seed consistency on multiple checkpoints of Stable Diffusion (we use checkpoints provided at `https://huggingface.co/CompVis/stable-diffusion` and `https://huggingface.co/runwayml/stable-diffusion-v1-5`) and find that images generated across these models with a fixed seed share significantly more visual similarity that those generated from independent random seeds (see Figure B.2.)

We take advantage of this property when evaluating the counterfactual impact of removing the training examples relevant to a given generated image (see Section 5.5). Specifically, we now expect that retraining a model on the full training set and then sampling from the same seed should produce a highly similar image to the generated image of interest. Thus, we can evaluate the counterfactual significance of removing the training examples with the top attribution scores for a given generated image by retraining and measuring the distance (in pixel space) of an image synthesized with the same seed to the original generated image.

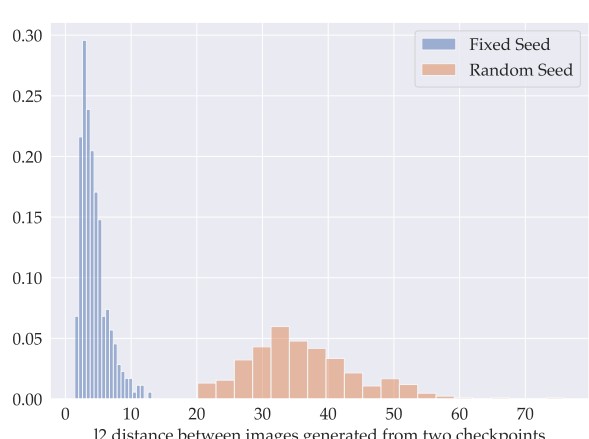
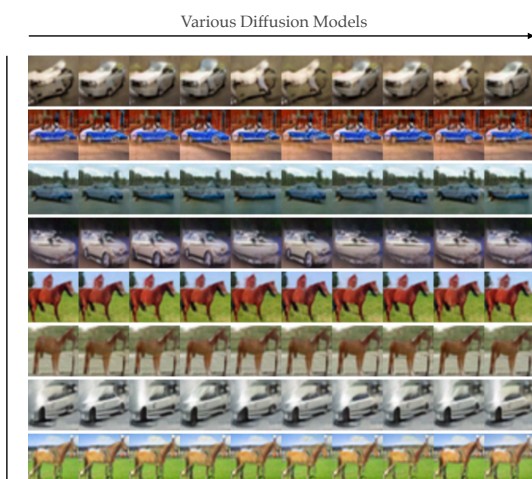

Figure B.1: **Seed consistency of CIFAR-10 DDPMs**. We find that across DDPMs trained independently on CIFAR-10, when using a fixed random seed during sampling, the resulting synthesized images are very similar, and often visually indistinguishable **(Right)**. Quantitatively, we find that the $\ell_2$ distance between images generated from two different models is significantly smaller when we fix the noise sequence **(Left)**.

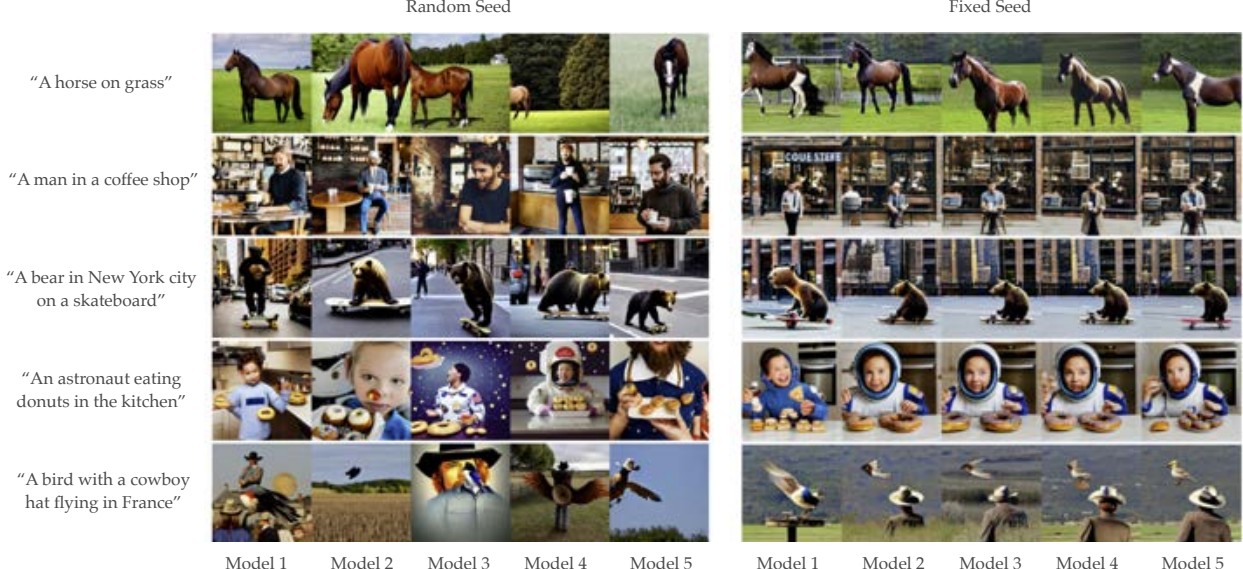

Figure B.2: **Seed consistency holds for Stable Diffusion models.** We find that seed consistency holds even for large, text conditioned model, specifically for Stable Diffusion models that are trained on LAION-5B. We compare multiple checkpoints of Stable Diffusion provided by Stability AI, and find that fixing the noise sequence during sampling surfaces very similar images (in comparison to using independent noising sequences).

## B.2 Attribution scores can drastically change over the course of the diffusion process

As additional motivation for performing attribution at individual steps rather than the entire diffusion trajectory, we highlight the following phenomena: *the same training image can be both positively influential and negatively influential for a generated sample at different steps.* For example, consider an image of a red car on a grey background generated by our DDPM trained on CIFAR-10 (See Figure B.3, top). We find that a specific training example of a red car on grass is the single most positively influential image according to Journey-TRAK at the early stages of the generative process (as it is forming the shape of the car), but is later the single most negatively influential image (possibly due to the difference in background, which could steer the model in a different direction). If we were to create an aggregate attribution score for the entire diffusion trajectory, it is unclear what the attribution score would signify for this training example.

To evaluate this phenomena quantitatively, we measure the percentage of generated images for which, for a given $K$, there exists a training example that is one of the top $K$ highest scoring images at some step and one of the top $K$ lowest scoring images at another step (according to Journey-TRAK). In Figure B.4, we show how this percentage varies with $K$. As a baseline, we also include the probability of such a training example existing given completely random attribution scores. We find that our observed probabilities match those expected with random scores, signifying that an image being highly positively influential at a given step *does not* decrease the probability that it is highly negatively influential at a different step.

To more broadly analyze the relationship between attributions at different steps, we additionally measure the Spearman's rank correlation (Spearman, 1904) between attribution scores for the same generated sample at different steps (see Figure B.5). We find that for steps that are sufficiently far from each other (around 500 steps), the attribution scores are nearly uncorrelated.

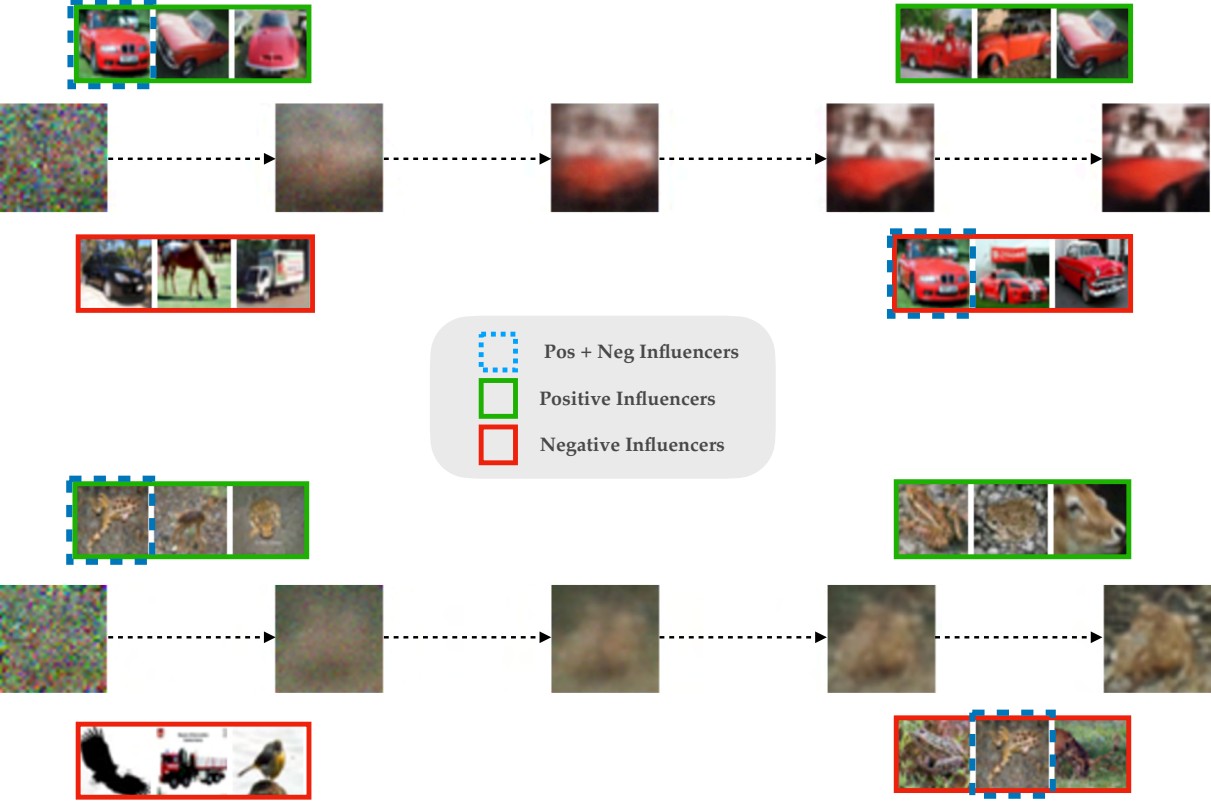

Figure B.3: **Overlap between positive and negative influencers.** Here, we visualize the generative process for two images generated by a DDPM on CIFAR for which there exists a training image that is both positively and negatively influential at different steps. If we consider an aggregate attribution score across all time-steps of the diffusion trajectory, we might lose the significance of such training examples which alternate between being positively and negatively influential during the sampling process.

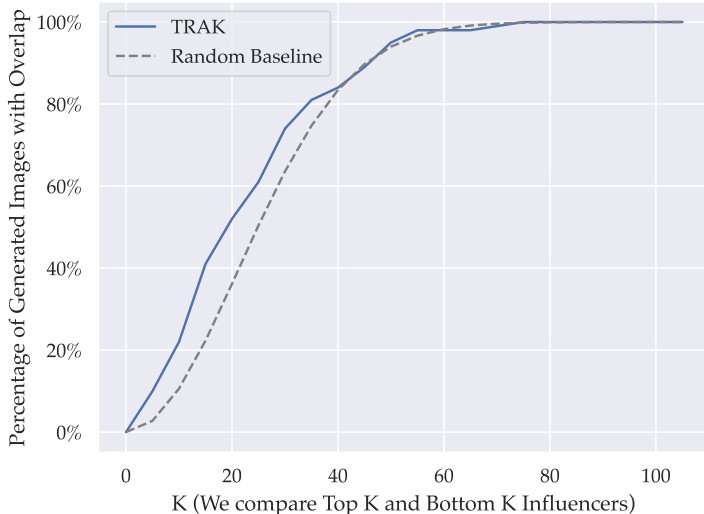

Figure B.4: **The relationship between positive and negative influencers.** Here, we plot the probability that within the attribution scores for a given generated image, there exists a training example that is one of the $K$ most positive influencers at some step and one of the bottom $K$ most negative influencers at another step. We compute this probability empirically with the attribution scores from Journey-TRAK and find that it closely aligns with the hypothetical baseline of completely random attribution scores. This signifies that being a top positive influencer at some step does not decrease the likelihood of being a top negative influencer at a different step.

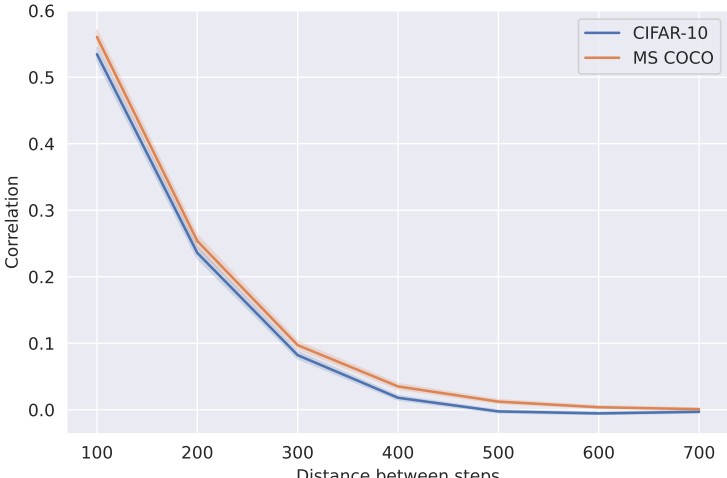

Figure B.5: **Correlation between attribution scores over steps.** Here, we plot the Spearman's rank correlation (Spearman, 1904) between the attribution scores for a given image generated by either our CIFAR-10 or MS COCO models at different steps, as a function of the distance between steps (results are averaged over 100 generated samples). As expected, steps that are closer in proximity have more closely aligned attribution scores. Interestingly, when we compute attributions at steps of distance 500 or more apart, the resulting scores are nearly uncorrelated.

### B.3 Feature analysis for Stable Diffusion

We analyze how the likelihood of different features in the final image varies over steps for images generated by a Stable Diffusion model,[6] similarly as we did for CIFAR-10 in Figure 3. In Figure B.6, we analyze an image generated using the prompt, *"A woman sitting on a unique chair beside a vase."* To measure the relative likelihood between two features (e.g., "white blouse" vs. "dark blouse"), we use a pre-trained CLIP model and measure whether the CLIP embedding of the generated image is closer to the text embedding of the first feature or the second feature. We sample 60 images at each step and report the average likelihood. We use 300 denoising steps to speed up the generation.

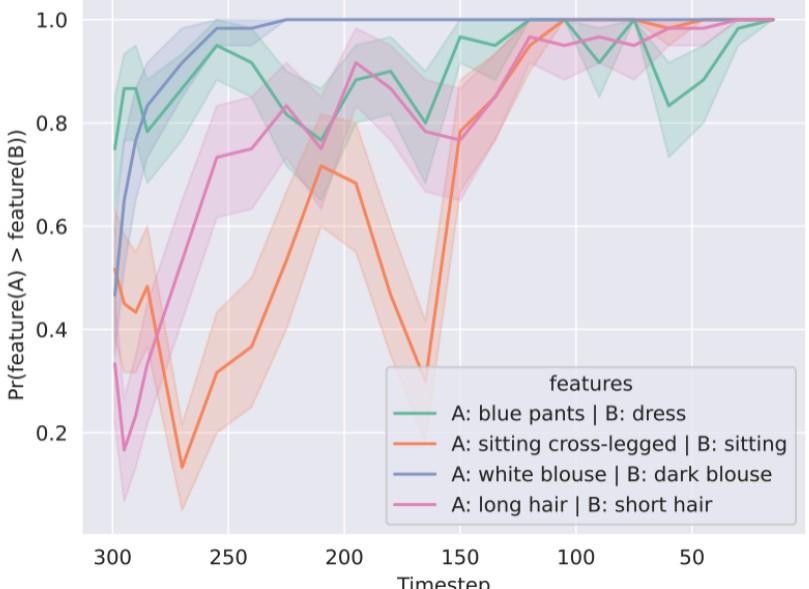
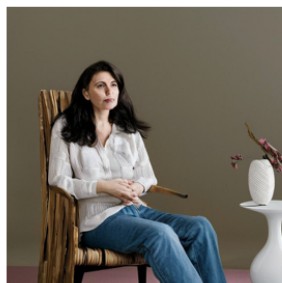

target image

Figure B.6: **Features appear at specific steps for Stable Diffusion. (Left)** For each pair of features, we plot the evolution in the relative likelihood of the two features (according to CLIP text-image similarity) in the conditional distribution $p_\theta(\cdot|\mathbf{x}_t)$. Features differ in when they appear, but usually rapidly appear within a short interval of steps. **(Right)** The generated image $\mathbf{x}_0$, sampled using $T = 300$ denoising steps.

---

[6]We use the `stabilityai/stable-diffusion-2` pre-trained checkpoint.

## C  Additional Related Work

**Data attribution.**   A long line of work has studied the problem of training data attribution, or tracing model behavior back to training data; we focus here on works done in the context of modern machine learning algorithms. Prior approaches include those based on the influence function and its variants (Hampel et al., 2011; Wojnowicz et al., 2016; Koh & Liang, 2017; Basu et al., 2019; Khanna et al., 2019; Achille et al., 2021; Schioppa et al., 2022; Bae et al., 2022), sampling-based methods that leverage models trained on different subsets of data (Ghorbani & Zou, 2019; Jia et al., 2019; Feldman & Zhang, 2020; Ilyas et al., 2022; Lin et al., 2022), and various other heuristic approaches (Yeh et al., 2018; Pruthi et al., 2020). These methods generally exhibit a strong tradeoff between predictiveness or effectiveness and computational efficiency Jia et al. (2021). The recent method of Park et al. (2023) significantly improves upon these tradeoffs by leveraging the empirical kernel structure of differentiable models. While most prior work primarily focus on the supervised setting, more recent works study attribution in generative settings, including to language models (Park et al., 2023) and to diffusion models (Wang et al., 2023). Concurrently to this work, Zheng et al. (2023) use Journey-TRAK to attribute diffusion models, but rely on heuristic design choices to design $f_t$. In addition, Dai & Gifford (2023) employ ensembling to attribute diffusion models.

In a recent work, Wang et al. (2023) propose a method for *efficiently evaluating* data attribution methods for generative models by creating custom datasets with known ground-truth attributions.

**Memorization in generative models.**   We can view *memorization* as a special case of data attribution where only few, nearly identical images in the training set are responsible for the generation of a corresponding image. Prior to the increasing popularity of diffusion models, a number of previous works studied memorization in other generative models. For example, Feng et al. (2021) study the impact of properties of a dataset (size, complexity) on training data replication in Generative Adversarial Networks (GANs), and van den Burg & Williams (2021) introduce a memorization score for Variational Autoencoders (VAEs) that can be additionally applied to arbitrary generative models. Following the release of large text-to-image diffusion models, the creators of one of these models (DALL · E 2) investigated memorization issues themselves and found that memorization could be significantly decreased through de-duplication of the training data (Nichol et al., 2022). Recently, Somepalli et al. (2022) explore the data replication behavior of diffusion models from the lens of "digital forgery," and identify many cases where, even when Stable Diffusion produces "unique" images, it directly copies style and semantic structure from individual images in the training set. On the other hand, Carlini et al. (2023) investigate memorization from the perspective of privacy, and show that query access to diffusion models can enable an adversary to directly extract the models' training data.

# D    TRAK details

In this section, given the close connection between our method Journey-TRAK and TRAK (Park et al., 2023), we provide a detailed description of the TRAK method.

Using our notation from Section 2.1, we have a learning algorithm $\mathcal{A}$ producing model parameters $\theta(S)$ when trained on a training set $S = (z_1, \ldots, z_n)$. Additionally, we define a model output function $f(z, \theta(S))$ of interest. Next, let use define a "feature map" $g : \mathcal{Z} \to \mathbb{R}^k$ as

$$g(z) := \mathbf{P}^\top \nabla_\theta f(z; \theta^\star), \tag{4}$$

i.e., a function taking an example $z$ to its corresponding gradient with respect to $\theta$ of the model output function $f$, projected with a random matrix $\mathbf{P} \sim \mathcal{N}(0,1)^{p \times k}$ for $k \ll p$. Finally, for brevity, let use define $G = [g(z_1), \ldots, g(z_n)]$ for all $z_i$ in the training set $S$.

Park et al. (2023) consider supervised learning settings (e.g., image classification). In short, they develop a method that attributes *scalar* output functions $f$. By large, they consider the "classification margin" output function, defined as $\log\left(\frac{p}{1-p}\right)$ where $p$ is the classfier's softmax probability of the correct class.

It turns out that for the above choice of model output function, one can write the attribution scores for a target sample $z$ as

$$\tau(z, S) := g(z)^\top (G^\top G)^{-1} G^\top \mathbf{Q}, \tag{5}$$

for a diagonal matrix $Q$ defined as $\mathbf{Q} = \mathrm{diag}\left(\left\{(1 + \exp(y_i \cdot f(z_i; \theta^\star)))^{-1}\right\}\right)$.

While convenient, this notation obfuscates the decomposition of this estimator into "change in parameters" and "change in output" which we outlined in Section 2.1 and Section 4. In particular, another way to write the estimator from Equation (5) is to define

$$\phi(z) := \mathbf{P}^\top \nabla_\theta L(z, \theta^\star), \tag{6}$$

where $L$ is the training loss. And again, we can define

$$\Phi = [\phi(z_1), \ldots, \phi(z_n)]. \tag{7}$$

With this new notation, we can write Equation (5) as

$$\tau(z, S) := g(z)^\top (G^\top G)^{-1} \Phi^\top. \tag{8}$$

Note that $(G^\top G)^{-1}$ remains intact, which is in divergence with the influence function derivations outlined in Pregibon (1981). This is due to an ablation study performed in Park et al. (2023).

## D.1    Differences between trak and Journey-trak

In Journey-TRAK, we do *not* use the simplification in Equation (8) and instead implement an estimator which follows Pregibon (1981) more closely:

$$\tau(z, S) := g(z)^\top (\Phi^\top \Phi)^{-1} \Phi^\top, \tag{9}$$

as expressed on line 17 in Algorithm 1. This is critical, since unlike in the classification setting, we can no longer make a simple decomposition similar to Equation (5), and using this simplified estimator would result in an overly biased estimator.

A key innovation in Park et al. (2023) is the reduction of the multi-class classification problems to binary classification via the "classification margin" output function. This choice makes the TRAK estimator much more tractable, especially when the number of classes is large (as in, e.g., ImageNet (Deng et al., 2009)). Similarly, when attributing steps of the diffusion process, "natural" output functions $f_t$ like the latent $\mathbf{x}_t$ itself are also high-dimensional. Thus, one of the key design choices for Journey-TRAK is the model output function $f_t$, which is diffusion-specific.

Another way to observe the key role that $f_t$ plays is to analyze the concurrent work of Zheng et al. (2023). In their paper, they adapt Journey-TRAK to develop a *timestep-global* attribution method by only making a change to $f_t$.

# E  Omitted plots

In this section, we present additional visualizations extending upon the figures in the main text. In Figure E.1 and Figure E.2, we visualize the most influential training examples identified by Journey-TRAK for a sample generated with a DDPM trained on CIFAR-10 and a LDM trained on MS COCO, respectively. In Figure E.3, we more concisely display attributions for additional samples generated by a CIFAR-10 DDPM. Finally, in Figure E.4 we display additional examples of the appearance of features over steps, and confirm that our findings in the main text hold across when different classification models are used for identifying a given feature.

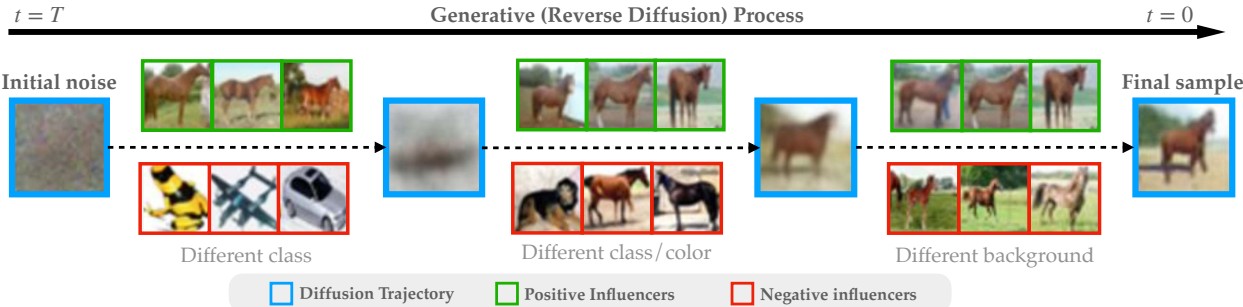

Figure E.1: An example of step-dependent attribution scores for a sample generated by a DDPM trained on CIFAR-10. At each step $t$, Journey-TRAK pinpoints the training examples with the highest influence (positive in green, negative in red) on the generative process at this step. In particular, positive influencers guide the trajectory towards the final sample, while negative influencers guide the trajectory away from it.

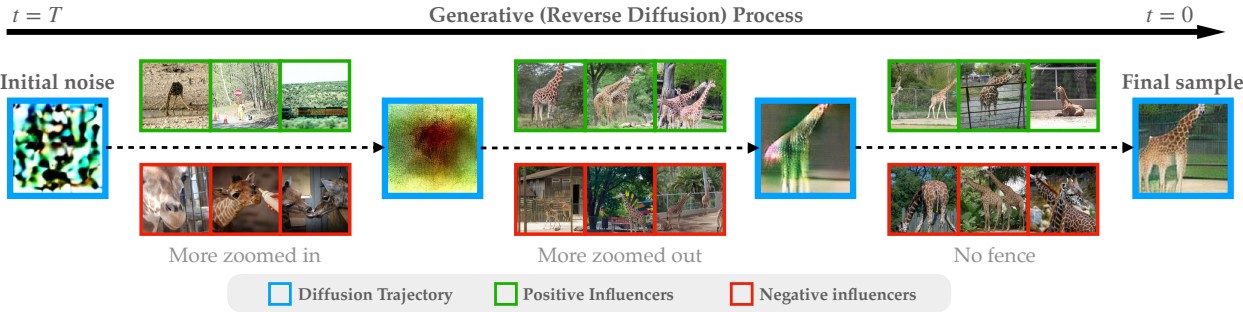

Figure E.2: An additional example of step-dependent attribution scores for a sample generated by a LDM trained on MS COCO. At each step $t$, Journey-TRAK pinpoints the training examples with the highest influence (positive in green, negative in red) on the generative process at this step. In particular, positive influencers guide the trajectory towards the final sample, while negative influencers guide the trajectory away from it.

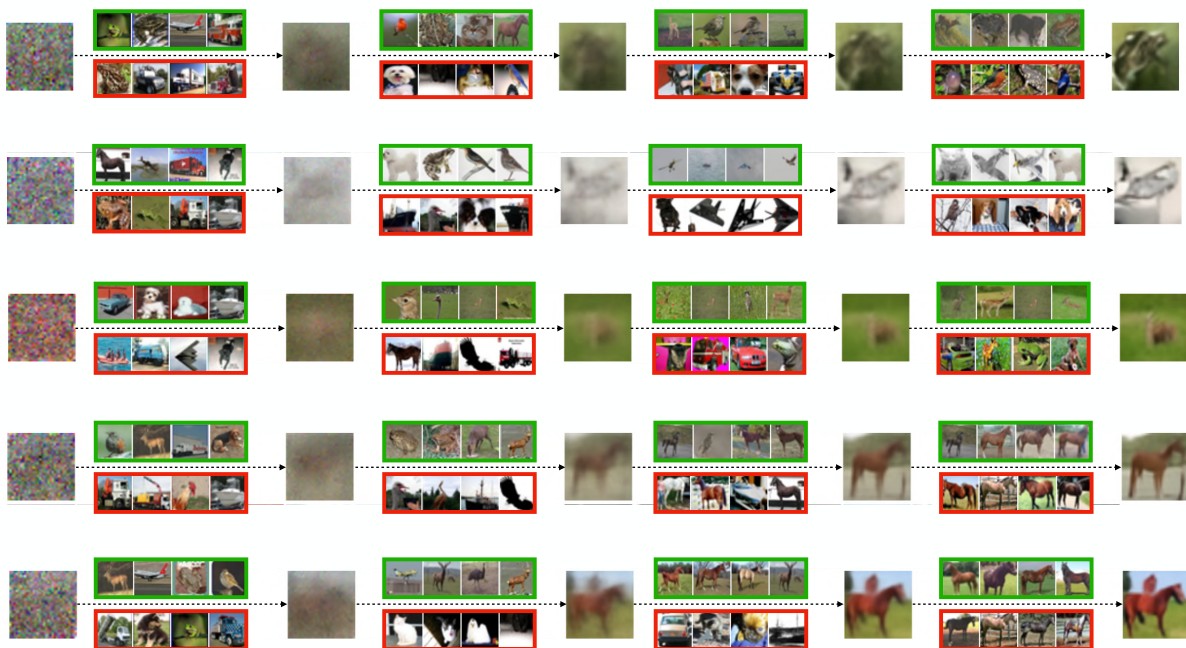

Figure E.3: Additional examples our attributions identified by Journey-TRAK. Here, we visualize the diffusion trajectory for generated images along with the most positively (green) and negatively (red) influential images at individual steps throughout the diffusion trajectory.

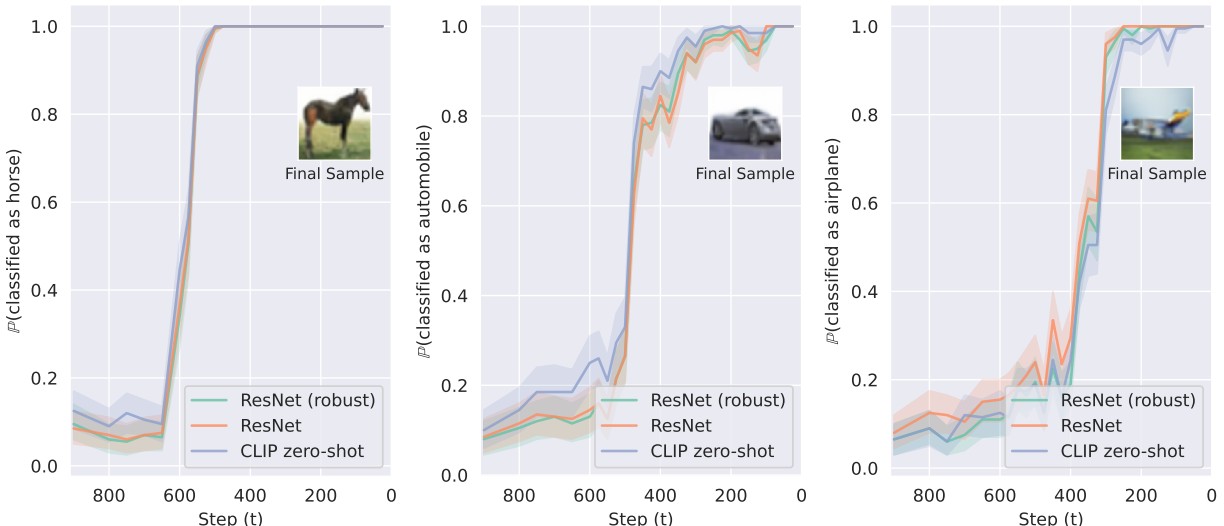

Figure E.4: Additional examples of the appearance of features over steps, similar to the analysis in Figure 3. In each plot, we show the likelihood that a sample generated from the distribution $p_\theta(\cdot|\mathbf{x}_t)$ contains a the feature of interest (in this case, the CIFAR-10 class of the final image) according to three different classifiers: a ResNet trained on the CIFAR-10 dataset with either standard or robust training, and zero-shot CLIP-H/14 model (Radford et al., 2021). Note that in each example, the likelihood that the final image contains the given feature increases rapidly in a short interval of steps, and that this phenomena is consistent across different classifiers.

