# OpenReview forum: "The Journey, Not the Destination: How Data Guides Diffusion Models"
_TMLR — Rejected by TMLR_

### Review · Reviewer_LGNq · 2024-07-06

**Summary Of Contributions:**

This work studies the problem of data attribution in diffusion models. The question it seeks to answer is which datapoints in the training set have a causal effect on a generated sample.

It is an application of the TRAK data attribution method (Park et al., 2023) to diffusion models. The key observation, supported by some initial analysis, is that data attribution on a per-timestep basis is more useful and interesting than an aggregated metric. To do this, the authors propose a timestep-specific "model output function," the central input of TRAK, and use it to build a data attribution algorithm. By a couple of metrics, this method performs substantially better than naive approaches, and through examples the authors show that it is qualitatively informative.

**Audience:**

Yes

**Broader Impact Concerns:**

No broader impact concerns.

**Claims And Evidence:**

No

**Requested Changes:**

# Critical
- This work needs to better contextualize itself within the broader literature. I have two specific concerns to this end:
	- This work's method is heavily based on TRAK, a year-old work, but TRAK is barely explained throughout the paper except at a very high level. Given TRAK's complexity and pertinence here, it needs to be explained in much more technical detail. The lack of context in this right means that the contents of the actual method - Algorithm 1 - come out of left field.
	- This work is about data attribution for diffusion models, but its literature review on this topic amounts to a single sentence about Wang et al. (2023). A quick Google shows that there is recent work in the area [1, 4, 5], and all of this should be at least briefly discussed. Most importantly, [5] also bases itself off of TRAK - how is that work (concurrent though it may be) different from the present one?
- Following on from the first point, there are a couple of things I find unclear about the method itself.
	- No clear justification was given for the timestep-specific model-output function $f_t$. It is (up to a constant) the ELBO of the model at the one-step sample estimate.
        -  I understand that the point here is to get a time-dependent scalar, but why this specific one? Why not something more naive, like $\lVert\varepsilon_\theta(x_t, t)\rVert$?
        - Furthermore, it is mentioned that $\hat{\mathbf{x}}_0^t$ is being used here as a proxy for samples from $p(\cdot | \mathbf{x}_t)$. What exactly is the original quantity you're trying to compute here involving $p( \cdot | \mathbf{x}_t)$?  How does this approximation affect the quality of the method? (In particular, Figure 2 shows this approximation is poor for large timesteps, and operates on the assumption that $p(\cdot | \mathbf{x})$ is "unimodal".)
	- What parts of Algorithm 1 are pulled directly from TRAK and what parts are new here? In particular, line 17 looks different from line 15 of the TRAK algorithm in Park et al. (2023) in the way it combines the $f_t$ and $f_\text{train}$ gradients. Is this a conscious design choice?
- Using notation from the text, Algorithm 1 requires $MNk$ model evaluations to start and then $MTk$ model evaluations to compute attributions for a single datapoint. This is seemingly expensive; can you discuss the time complexity and/or report the amount of time the method takes for an individual datapoint?
- In Appendix C.1: "diffusion models are consistent across seeds", _seed consistency_,the propensity of diffusion models trained with different seeds to learn the same maps between latents and data, is framed as a theoretically unexpected property. This is actually not true; it is expected.
	- Diffusion models seek to learn the score over all timesteps of a fixed diffusion process; this score is unique. This is why, even if you train multiple a diffusion models even with different architectures, you will get very similar images starting from the same latent. In this sense, all diffusion models _do_ share the same latent space. (This is very different from eg. VAEs, GANs, and normalizing flows, whose optimal mappings from latents to data are not at all unique).
	- [2] has some interesting analysis on this.

# Non-Critical
- Ilyas et al. (2022)  on page 3 should be a citep.
- Missing period in the last sentence of section 3.1.
- The authors might consider discussing [3], which studies a very similar notion as Figures 3, C.3, and C.10, identifying a "phase transition" in the diffusion process wherein specific details of the image are decided.
- What's behind the decision not to use the evaluation methods of Wang et al. (2023)?
- Although the writing is clear, it could be better organized and more concise in places. A couple of organization examples:
	- The "Implementing Our Approach" section (page 6) partially describes $f_t$ without actually defining it. It is then actually defined 2 pages later. Why separate these parts?
	- The start of section 3.2 contains 7 lines of text giving partial definitions of and comparing the two evaluation metrics before their actual definitions are given below - it would be more economical to save the comparisons until afterwards.


# References

[1] Dai, Zheng, and David K. Gifford. "Training data attribution for diffusion models." _arXiv preprint arXiv:2306.02174_ (2023).

[2] Kadkhodaie, Zahra, et al. "Generalization in diffusion models arises from geometry-adaptive harmonic representations." _The Twelfth International Conference on Learning Representations_.

[3] Sclocchi, Antonio, Alessandro Favero, and Matthieu Wyart. "A phase transition in diffusion models reveals the hierarchical nature of data." _arXiv preprint arXiv:2402.16991_ (2024).

[4] Xie, Tong, et al. "Data Attribution for Diffusion Models: Timestep-induced Bias in Influence Estimation." _arXiv preprint arXiv:2401.09031_ (2024).

[5]  Zheng, Xiaosen, et al. "Intriguing Properties of Data Attribution on Diffusion Models." _The Twelfth International Conference on Learning Representations_.

**Strengths And Weaknesses:**

# Strengths
The method appears to be effective and versatile. It allows for patch-wise and/or timestep-wise attribution. While data attribution is a difficult task to evaluate, the proposed metrics are sensible and indicate the method is generally performing well (not just on cherry-picked examples).

I also enjoyed the analysis, which surfaces a number of interesting tidbits; to name a few:
- The probability of certain concepts appearing in the generated image experience unexpectedly sharp increases at specific points in the trajectory.
- The training examples with "negative influence" in the generation process become more similar to the generated sample as time nears 0.
- Nearest neighbors in pixel- or CLIP-space in the training set are generally NOT very relevant to a generated sample in attribution terms.

# Weaknesses
Please find the requested changes below.

---

### Review · Reviewer_JiSC · 2024-07-10

**Summary Of Contributions:**

This paper proposed a method to attribute image synthesized by diffusion models. The authors adapted TRAK, a previous data attribution method for language models, for the diffusion settings. It uses a step-specific approach to understand which training samples influence each diffusion step and how the influences evolve along the whole trajectory. Meanwhile, the proposed method allows to isolate attribution for different features in a image by masking unwanted patches.

**Audience:**

Yes

**Claims And Evidence:**

Yes

**Requested Changes:**

Requested Changes:

1. According to Related Work, there are other methods for attributing diffusion models. Are they comparable with the proposed method? If so, what are their performances?

2. How to understand Figure 4? In this figure, we can see TRAK has better LSD comparing to CLIP and pixel similarity. But as LSD is essentially Spearsman rank correlation, a value of 0.2~0.3 indicates very weak ranking power. What is an acceptable LSD value for data attribution purpose? How to interpret the pivot point in LSD curve along the trajectory (LSD increases initially and then decreases)?

3. In the Experiments section, TRAK is used to refer the proposed adaption for diffusion models. Simply calling it TRAK can be confusing, since it can also refers to the original TRAK method. It could be better to phrase it differently.

**Strengths And Weaknesses:**

Strength:

1. The proposed method adapts TRAK, an attribution method for language model, for diffusion models.

2. The proposed method is evaluated from multiple perspectives, including qualitative visual inspection, counterfactual validation, patch-based attribution, memorization analysis, and it achieves better performance comparing to CLIP and pixel similarity.


Weakness:

1. The baselines used in this paper are relatively simple and naive, namely CLIP similarity and pixel similarity.

2. Some of the descriptions are not clear enough and will be discussed in the “Requested Changes” section.

---

### Review · Reviewer_xGuF · 2024-07-14

**Summary Of Contributions:**

This work provides a data attribution framework for diffusion model, helping us identify those attributes during the diffusing process.

**Audience:**

Yes

**Claims And Evidence:**

Yes

**Requested Changes:**

It would be more clear if author have a comparison between the defined functions in diffusion model and those functions in the general framework.

**Strengths And Weaknesses:**

Strengths:
1. Presentation is clear and easy to follow.
2. Present a data attribution framework for diffusion models including defining data attributes, model output function and etc.
3. Provide two complementary metrics for evaluating attributions and use them to validate the proposed method on diffusion models.

Weaknesses:
1. This work leverages the TRAK framework to get attribute score of the diffusion models and uses the LDS for analysis. It seems to be very straightforward.

---

### Decision · Action_Editor_zbYi · 2024-08-20

**Recommendation:** Reject

**Comment:**

This paper proposed a method based on the TRAK method from language model to attribute image synthesized by diffusion models, which uses a step-specific approach to try to learn the influence and evolution of each training sample. All reviewers agree the method is interesting and on time. There are some concerns about the novelty and relation with TRAK, which were addressed in the rebuttal by adding more detailed discussions. However, there are still some other problems mentioned by the reviewers that were not fully addressed, one of which is the experiments. The current experiments compare the proposed method with two simple baselines, which is not enough to demonstrate the effectiveness and validation of the proposed method. The rebuttal did not fully address this comment, more baseline comparisons are expected to strengthen the paper.

To sum up, the problem is important and the proposed method is indeed interesting. However, given the lack of enough empirical evidence to support the effectiveness of the proposed method, I have to recommend rejection this round. If the authors can revise the paper and add more experiment comparisons, I am happy to oversee the paper again in the future submission.

**Audience:**

The paper will be found interested by researchers from diffusion models and data attribution.

**Claims And Evidence:**

The claims and evidence are partially supported by the evidence from the paper. According to the reviewers, I believe there is still room for improvement.

**Resubmission Of Major Revision:**

The authors may consider submitting a major revision at a later time.